# The onset of widespread marine red beds and the evolution of ferruginous oceans

Haijun Song [1], Ganqing Jiang [2], Simon W. Poulton[3], Paul B. Wignall[3], Jinnan Tong[1], Huyue Song[1], Zhihui An[1], Daoliang Chu[1], Li Tian[1], Zhenbing She[1] & Chengshan Wang[4]

Banded iron formations were a prevalent feature of marine sedimentation ~3.8–1.8 billion years ago and they provide key evidence for ferruginous oceans. The disappearance of banded iron formations at ~1.8 billion years ago was traditionally taken as evidence for the demise of ferruginous oceans, but recent geochemical studies show that ferruginous conditions persisted throughout the later Precambrian, and were even a feature of Phanerozoic ocean anoxic events. Here, to reconcile these observations, we track the evolution of oceanic Fe-concentrations by considering the temporal record of banded iron formations and marine red beds. We find that marine red beds are a prominent feature of the sedimentary record since the middle Ediacaran (~580 million years ago). Geochemical analyses and thermodynamic modelling reveal that marine red beds formed when deep-ocean Fe-concentrations were > 4 nM. By contrast, banded iron formations formed when Fe-concentrations were much higher (> 50 μM). Thus, the first widespread development of marine red beds constrains the timing of deep-ocean oxygenation.

[1] State Key Laboratory of Biogeology and Environmental Geology, School of Earth Science, China University of Geosciences, Wuhan 430074, China. [2] Department of Geoscience, University of Nevada, Las Vegas, NV 89154-4010, USA. [3] School of Earth and Environment, University of Leeds, Leeds LS2 9JT, UK. [4] State Key Laboratory of Biogeology and Environmental Geology, China University of Geosciences, Beijing 100083, China. Correspondence and requests for materials should be addressed to H.S. (email: haijun.song@aliyun.com)

Banded Iron Formations (BIFs) first appeared 3.85 billion years ago (Ga) in the Archean and were particularly prevalent around 2.6–2.4 Ga when they attained thicknesses of hundreds of metres[1, 2]. BIFs are composed predominantly of ferric and ferrous minerals, including hematite ($Fe_2O_3$), magnetite ($Fe_3O_4$), and siderite ($FeCO_3$)[1]. In spite of debate on the origin of these Fe-bearing minerals, including chemical precipitation under oxygen-free conditions[1], photo-oxidation by ultraviolet light[3], and microbial oxidation[4], it is clear that BIFs provide a prominent signature of anoxic, Fe-rich oceans early in Earth's history[5, 6].

With the exception of a temporally restricted episode of BIF deposition associated with possible 'snowball' Earth glaciations in the Neoproterozoic[7], BIFs ceased depositing at ~1.8 Ga[2]. This change has been used as evidence for a fundamental shift in ocean redox conditions, either to euxinic[8] or to oxygenated oceans[9]. Recently, however, iron geochemical studies have shown that ferruginous deep oceans were prevalent throughout the mid- to late-Proterozoic and may have persisted into the Cambrian[10–13], observations which are supported by redox-sensitive trace element data[14–18]. By contrast, euxinic conditions appear to have been limited to mid-depth waters on productive continental shelves or near shelf margins[6, 12, 19].

The phanerozoic iron speciation data suggests that ferruginous conditions also occurred in certain localities, often repeatedly, during oceanic anoxic events (OAEs), such as those in the Arabian Margin during the Early Triassic[20], the subtropical shelf of Morocco during the Cretaceous OAE2[21, 22], and the central Atlantic realm (Demerara Rise) during the Cretaceous OAE3[22]. However, the general absence of BIFs after ~1.8 Ga raises the question of how subsequent ferruginous episodes differed from those earlier in Earth history.

To address this question, we first provide a new record of the distribution of marine red beds (MRBs) through time. Red beds are common in sedimentary successions, but most (termed continental red beds) formed in terrestrial settings where the colouring agent (hematite) was developed under oxygenated atmospheric conditions[23]. More recently, however, MRBs have been documented in Cretaceous strata of worldwide extent[24–26]. Here we show that MRBs were geographically widespread beyond the Cretaceous, and occurred during sporadic intervals from the middle Ediacaran and throughout the Phanerozoic. We subsequently consider how the genesis of MRBs reflects changes in ocean chemistry after the disappearance of BIFs, and use this information to estimate secular changes in $Fe^{2+}$ concentrations in the ocean throughout Earth history.

## Results

**Traits and geological settings of MRBs.** The first globally distributed MRBs appear in middle Ediacaran (~580 million years ago) strata (Figs. 1e, 2a, 3i). Characterized by pink-red limestones, dolostones and shales that range from <1 to 40 m thick, the middle Ediacaran MRBs have iron concentrations of <3 wt% (Supplementary Table 1), which is much lower than BIF values but high enough to impart a characteristic red colouration. Both palaeolatitude and geographical extent (Figs. 2a, 3j) show that Ediacaran and Phanerozoic MRBs are not localised phenomenon, but rather, their distribution is as widespread as BIFs. We document a total of five global MRB intervals from the Phanerozoic, including Cambrian, Late Devonian, Early Triassic, Jurassic

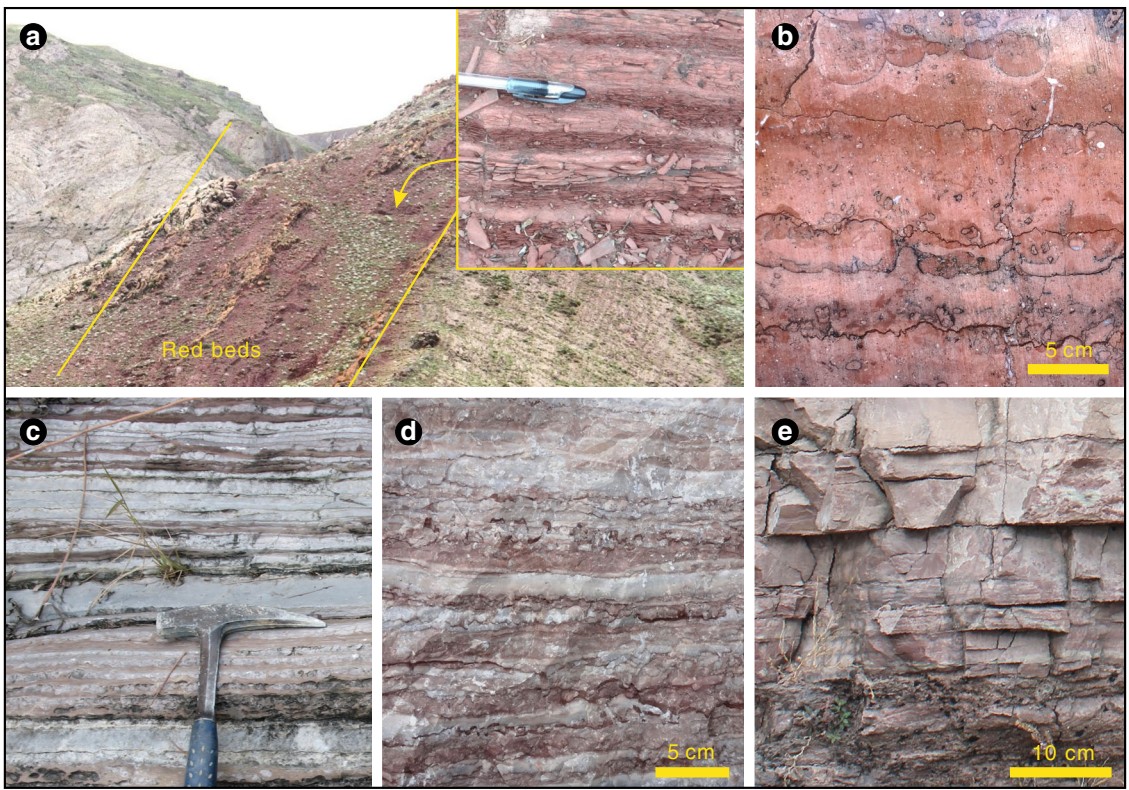

**Fig. 1** Representative marine red beds from Ediacaran and Phanerozoic successions. **a** Late Cretaceous mudstone interbedded with shale (Chuangde Formation, Tibet) containing abundant plankton foraminifers, indicating a pelagic facies. **b** Early Jurassic limestone (Adnet Formation, Austria) with abundant ammonites, suggesting a deep water environment. **c** Early Triassic, interbedded *grey* and *red* limestones (offshore facies) from the Luolou Formation, South China. **d** Late Devonian *grey* and *red* limestone (offshore facies) from the Wuzhishan Formation, South China. **e** Middle Ediacaran dolostone interbedded with silty shale from Krol B, Lesser Himalaya, India

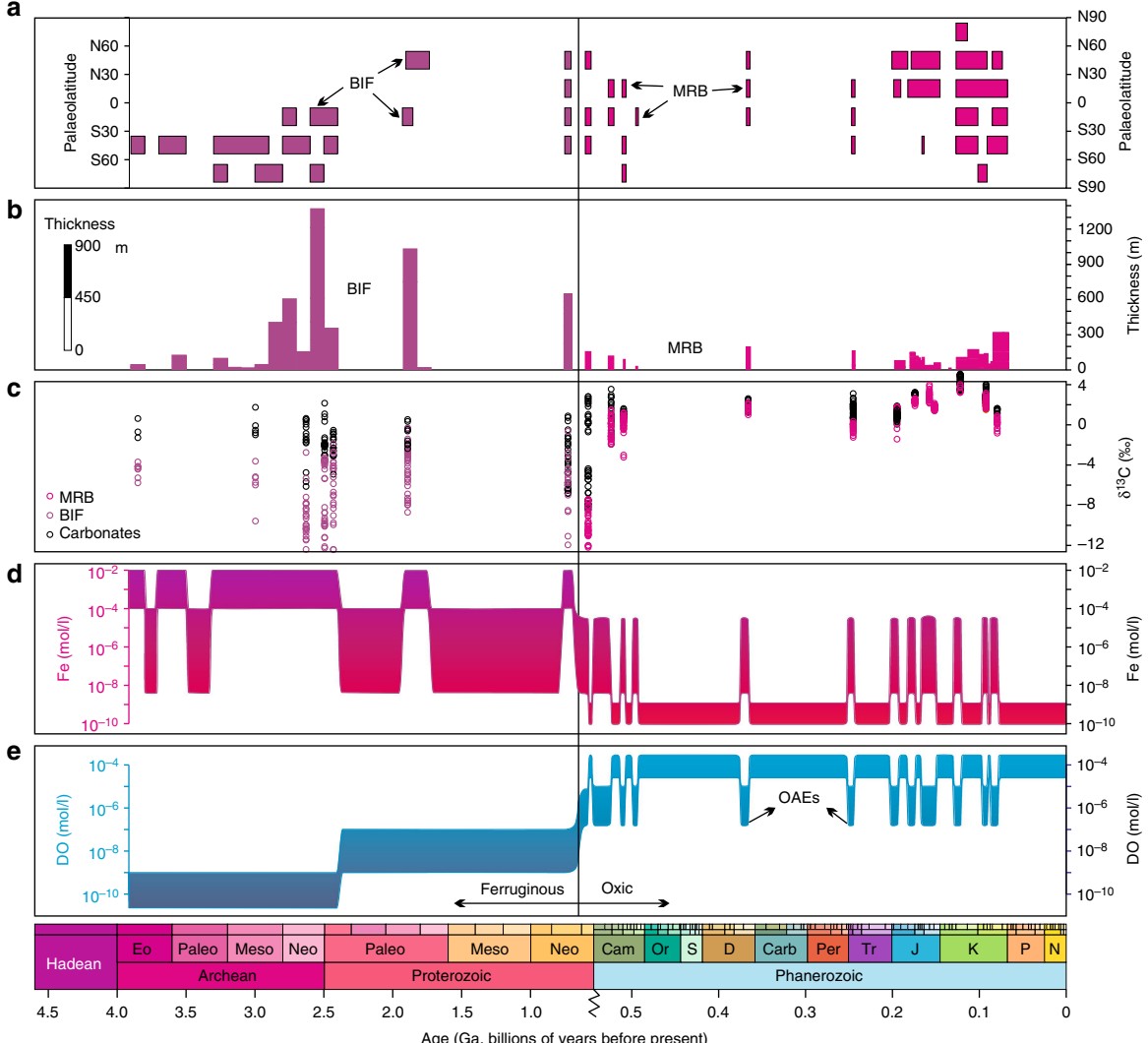

**Fig. 2** Secular distributions and carbon isotopes of Fe-related rocks and the evolution of marine redox and iron states. **a** Palaeolatitudinal distribution of banded iron formations (BIFs) and marine red beds (MRBs). The data are compiled from Supplementary Table 2. Note that the palaeolatitude data are at a lower confidence level for Archean and early Proterozoic interval. **b** Temporal distribution of BIFs and MRBs with thickness information (see data in Supplementary Figs. 1, 2 and Supplementary Table 2). **c** Carbon isotopes in iron-related sedimentary rocks and adjacent carbonates (*purple circle* are the data from banded iron formations, *magenta circles* are the data from red beds, *black circles* are the data from adjacent *grey* carbonates; see data in Fig. 3 and Supplementary Figs. 3–7 and Supplementary Tables 1, 3). **d** Evolution of deep-water (below storm wave base) iron concentrations based on numerical model (Fig. 4) and modern analogues. The lower limit of iron concentrations for BIF and MRB are 50 μM and 4 nM, respectively (see discussion in text). **e** Evolution of deep ocean (below storm wave base) redox states in the Phanerozoic and Precambrian (after refs [9, 57])

and Cretaceous episodes (Figs. 1 and 2). These Phanerozoic MRBs consist mainly of red carbonate (Fig. 1b–e) and red mudstone (Fig. 1a) that sometimes alternate with grey carbonate (Fig. 1c, d). Iron contents are <1% in carbonate and 1–6% in mudstone, which are only slightly elevated compared to adjacent rocks (Supplementary Table 1). Sedimentological, petrographic and mineralogical analyses of the Cretaceous MRBs indicate that nanometer-scale hematite and goethite (mostly transferred to hematite during late diagenesis), instead of detrital iron, are the major colour agents of MRBs[25–27]. Similar to those documented from Cretaceous MRBs[25, 26], high $Fe^{3+}/Fe^{2+}$ ratios are observed in Early Triassic MRBs (Supplementary Table 1).

MRBs are encountered in a broad range of depositional settings, spanning the entire spectrum from nearshore to deep basin environments. Ediacaran MRBs are commonly developed as oolitic and stromatolitic carbonates in shallow-water settings (Supplementary Note 1), but in the Doushantuo Formation of South China and the Krol B of northern India, red dolostone and

limestone are interbedded with laminated shales, suggesting offshore occurrences. Most Phanerozoic MRBs were formed in deeper offshore facies. For example, in the Early Triassic of the Nanpanjiang Basin of South China, MRBs occur in slope and basin settings but only rarely in shallow-water carbonate platforms. In contrast, Cretaceous MRBs have a wider range, from offshore slope to pelagic basin, although they are also encountered in some shallow platform sections. The more common record of deep-water MRBs in younger (e.g., Mesozoic) successions is likely due to better preservation of unsubducted slope-basin sediments.

**Coupling between MRBs and OAEs.** Each of the MRB events we have identified follows a period of ocean anoxia. Middle Ediacaran MRBs occurred synchronously with deep-water oxygenation after the Gaskiers glaciation[10]. Cambrian MRBs occurred in the aftermath of early, middle and late Cambrian OAEs[28–30]. Early Famennian (Late Devonian) MRBs appeared during the

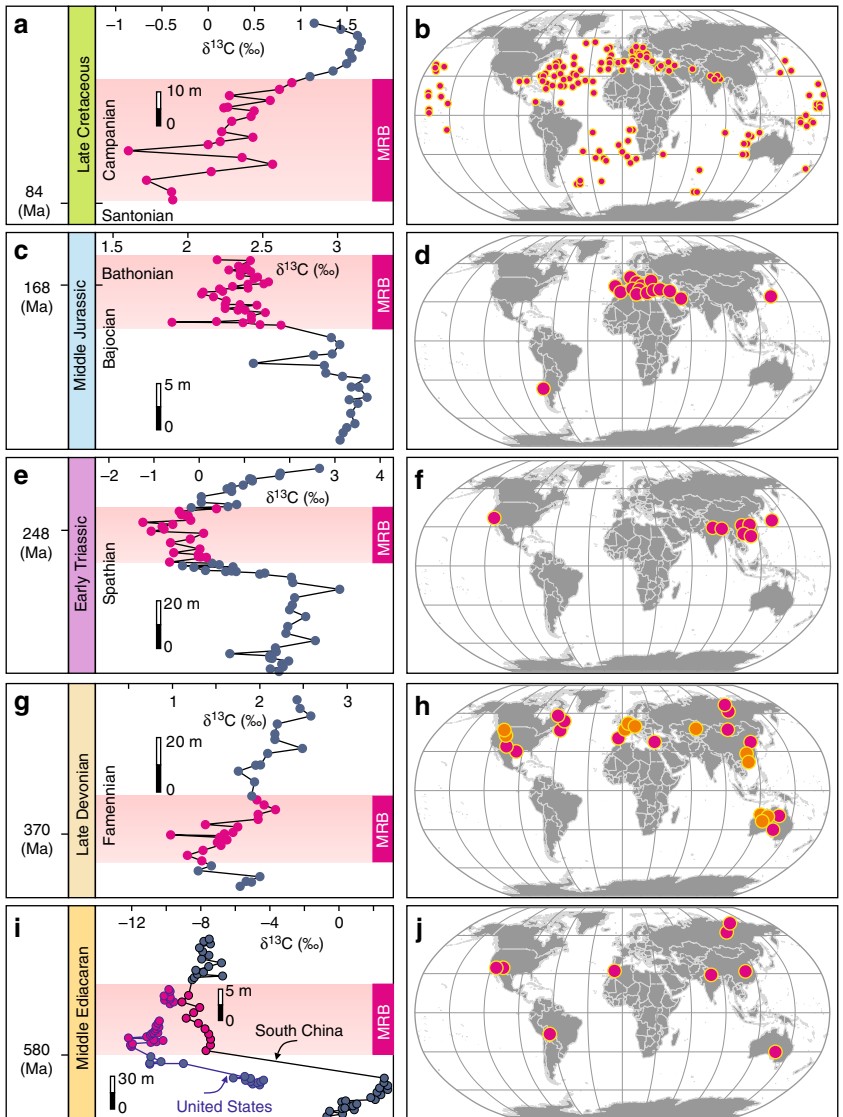

**Fig. 3** Carbonate carbon isotope curves and the distribution of marine red beds. **a** Carbonate $\delta^{13}C$ curve from the Late Cretaceous strata in Chuangde, Tibet, China. **b** Spatial distribution of Cretaceous MRBs. **c** Carbonate $\delta^{13}C$ curve from the Middle Jurassic strata in Puerto Escaño, southern Spain[41]. **d** Spatial distribution of Jurassic MRBs. **e** Carbonate $\delta^{13}C$ curve from the Spathian (Early Triassic) in Mingtang, South China. **f** Spatial distribution of Early Triassic MRBs. **g** Carbonate $\delta^{13}C$ curve from the Famennian (Late Devonian) strata in Baisha, South China. **h** Spatial distribution of Late Devonian (*magenta*) and Cambrian (*orange*) MRBs. **i** Carbonate $\delta^{13}C$ curves from the middle Ediacaran in Shijiahe, South China and northern Mesquite Mountains, United States. **j** Spatial distribution of Ediacaran MRBs

termination of the Frasnian–Famennian boundary OAE[31]. Early Triassic MRBs occurred following the Permian–Triassic boundary OAE[32]. Jurassic MRBs appeared right after the regional anoxia event at the Triassic–Jurassic boundary[33] and the Toarcian OAE[34], respectively, although sporadic red beds were also reported from Middle-Late Jurassic strata of the Tethyan regions[35, 36]. With few exceptions[37], Cretaceous MRBs follow immediately after middle-late Cretaceous OAEs[26, 38], including the Aptian–Albian OAE, Cenomanian-Turonian OAE, and Santonian-Campanian OAE. Red beds have also been reported from Early Silurian (Telychian) successions[39, 40], but they are mostly distributed surrounding uplifted "old lands" and are likely of detrital origin[40] and so are not included in the compilation (Supplementary Note 2).

**Coupling between MRBs and negative carbon isotope shifts.** We measured carbonate carbon isotopes ($\delta^{13}C$) in MRBs and stratigraphically adjacent rocks from the Ediacaran

and some Phanerozoic strata of North America, South China, and Tibet (Supplementary Figs. 3–7). Negative $\delta^{13}C$ shifts are identified in all newly-studied MRB intervals (Fig. 3). The average $\delta^{13}C$ gradients between MRBs and adjacent rocks are −8.74‰, −0.44‰, −1.53‰, and −1.25‰ for the middle Ediacaran, Late Devonian, Early Triassic, and Late Cretaceous, respectively (Fig. 3). A similar $\delta^{13}C$ signal has also been reported in a Middle Jurassic MRB from southern Spain[41]. The magnitude of $\delta^{13}C$ shift during the Ediacaran MRB is much larger than those of the Phanerozoic examples. This large $\delta^{13}C$ shift, the Shuram carbon isotope excursion, has been reported globally, but its origin remains debated (see Supplementary Note 3). Considering that low $\delta^{13}C$ values down to −12‰ were also present in diagenetic carbonates in BIFs (Fig. 2c), it is conceivable that diagenetic oxidation of organic carbon, including iron reduction (using iron oxides as electron acceptors) contributed, at least, to the heterogeneity of the Shuram $\delta^{13}C$ excursion.

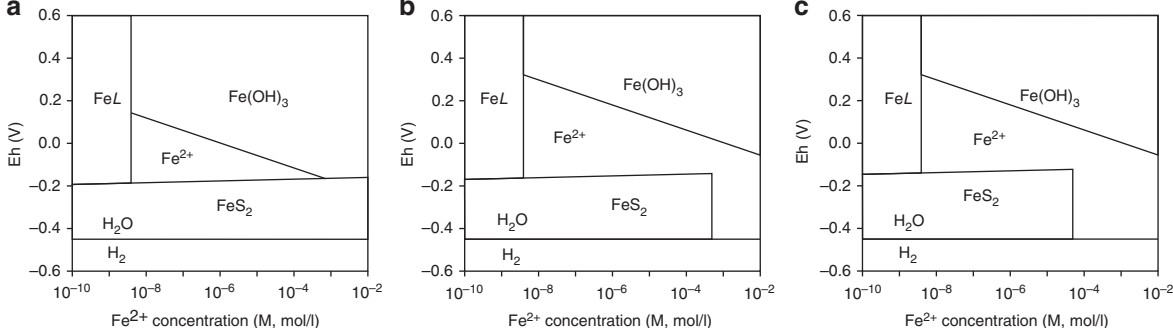

**Fig. 4** Thermodynamic models interpreting required ferrous ion concentrations for the formation of BIF and MRB. **a** Graph of Eh vs. $Fe^{2+}$ concentration for Phanerozoic oceans at $T = 25\,°C$ and $P = 1\,bar$. The calculations are based on $mSO_4^{2-} = 15\,mM$, and pH = 7.5. **b** Graph of Eh vs. $Fe^{2+}$ concentration for Proterozoic oceans at $T = 25\,°C$ and $P = 1\,bar$. The calculations are based on $mSO_4^{2-} = \sim1\,mM$, and pH = 7.0. **c** Graph of Eh vs. $Fe^{2+}$ concentration for Archean oceans at $T = 25\,°C$ and $P = 1\,bar$. The calculations are based on $mSO_4^{2-} = \sim100\,\mu M$, and pH = 6.5. The sulfate and pH data used here are from refs [45, 46, 58, 59]. Fe$L$: iron-binding ligands. The pH values of ancient seawater are consistent with the recent estimates of $\sim7.5$–9 for Phanerozoic and $\sim6.5$–7.0 for Archean and Proterozoic[60]

## Discussion

Two different hypotheses have been employed to explain the origin of Cretaceous and Jurassic MRBs: microbial-induced iron oxidation during sedimentation[42] and iron oxidation under oxygenated, oligotrophic conditions[26]. Our finding of widely distributed MRBs following global ocean anoxia suggests that anoxic, ferruginous water column conditions were the prerequisite for MRB formation. We propose that displacement of $Fe^{2+}$-rich anoxic waters into oxic waters during and following the OAEs led to precipitation of unstable, poorly crystalline hydrous ferric oxide phases that subsequently aged to hematite.

In the modern oxic ocean, iron concentrations are extremely low ($\sim0.1$–1 nM) and most 'dissolved' iron is bound to organic molecules as iron-binding ligands (Fe$L$). The average upper limit of Fe ligand concentration in modern oceans is about 4 nM[43]. In this case, authigenic Fe oxides only form when the dissolved $Fe^{2+}$ concentration is up to 4 nM. Because iron binding to ligands is an equilibrium process, which means that more iron would be tied up in ligands at higher dissolved iron concentrations[44], the actual Fe concentrations required for the formation of MRBs may be higher than 4 nM. We ascribe the extra dissolved $Fe^{2+}$ (higher than the modern ocean Fe concentration of 0.1–1 nM) required for MRB formation to the build-up of water-column $Fe^{2+}$ under anoxic oceanic conditions.

Both BIF and MRB record oxidation of reduced iron in the ocean, but their required water-column $Fe^{2+}$ concentrations vary significantly. Iron concentrations in BIFs are generally >20 wt%[1], and petrological and experimental evidence shows that the iron oxide and carbonate minerals in BIFs may not be primary precipitates, but products of post-depositional alteration of precursor ferric hydroxides ($Fe(OH)_3$)[2]. BIFs are commonly massive, up to hundreds of metres thick, while MRBs are a few metres to tens of metres thick (Fig. 2b, Supplementary Table 2). Many BIFs in Archean and Paleoproterozoic form giant iron ores with iron-rich deposition over 10,000 billion tonnes[2]. Thermodynamic modelling indicates that the minimum value of $mFe^{2+}$ (dissolved $Fe^{2+}$ concentration) required for deposition of $Fe(OH)_3$ when Eh <−0.16 is >50 μM (Fig. 4). In this case, the lower limit of $mFe^{2+}$ for BIF to form is $\sim50$ μM. When iron concentrations are below this lower limit (50 μM), iron is easily exhausted by reaction with dissolved sulphide produced by sulphate reducing bacteria, even under the much lower sulphate concentrations of <100 μM estimated for Archean oceans[45–47]. This lower $mFe^{2+}$ limit is close to previous estimates of $\sim54$ μM[48] and 40–120 μM[49], based on calculations assuming seawater saturation with respect to

siderite and calcite. Alternatively, ferric hydroxides could have been formed by microbial oxidation[4]. Experimental studies show that microbial oxidation rates increase substantially when the $Fe^{2+}$ concentration rises from 2 to 4 mM, suggesting that bacterial precipitation of ferric hydroxides also requires relatively high $Fe^{2+}$ concentrations[4]. Together, these constraints suggest that for BIFs to form, iron concentrations were likely >50 μM, a value that is several orders higher than that required for MRB formation.

Although BIFs are characteristic of the Precambrian, there is a big gap in their occurrence in the mid-Proterozoic ($\sim1.8$ to $\sim0.8$ Ga, see Fig. 2a). This is likely due to a decrease in dissolved $Fe^{2+}$ concentrations, potentially coupled with removal of $Fe^{2+}$ as pyrite during upwelling onto euxinic continental shelves[8, 12]. A decrease in dissolved $Fe^{2+}$ is also consistent with the development of the supercontinent Columbia during the 1.8–1.3 Ga period, which underwent only minor modifications to form the next supercontinent Rodinia at 1.1–0.9 Ga[50]. The unusually quiescent state of global tectonics during the mid-Proterozoic may have resulted in reduced hydrothermal iron flux, leading to oceanic iron concentrations lower than that required for BIF precipitation. Although it is difficult, if possible, to precisely quantify dissolved $Fe^{2+}$ concentrations during the mid-Proterozoic, $Fe^{2+}$ concentrations during this period may be comparable with or higher than that required for MRB formation (>4 nM), but low oxygen content in atmosphere and shallow oceans[51, 52] may have limited the formation of red beds to terrestrial and localised nearshore environments.

The onset of widespread MRBs during the middle Ediacaran may be a marker for substantial change in ocean chemistry. Numerical modelling suggests that for BIFs to form, deep-ocean Fe concentrations were likely higher than 50 μM, while MRBs require much lower dissolved Fe concentrations (>4 nM). The similarity of the iron cycle between Ediacaran and Phanerozoic MRBs suggests that anoxic Ediacaran oceans were more comparable to Phanerozoic anoxic oceans rather than the strongly ferruginous oceans of the Archean and early Proterozoic. At other times in the Phanerozoic, iron concentrations were much lower, which precluded the formation of MRBs. However, we estimate iron concentrations during the formation of MRBs to be in a similar range (possibly at the lower end) to those of the mid-Proterozoic, where MRBs did not form (Fig. 2). We suggest that this apparent contradiction is a consequence of the transition to widespread deep ocean oxygenation in the terminal Proterozoic[10, 53, 54], which promoted the formation of MRBs in the aftermath of periods of ocean anoxia. MRBs are thus evidence

for anoxic episodes occurring during long-term intervals of deep ocean oxygenation and their appearance in the middle Ediacaran constrains the timing of deep-ocean ventilation.

## Methods

**Carbon isotope and iron chemistry analyses**. The data are presented in Supplementary Table 1. MRB and BIF samples selected for geochemical analyses include both drill core and hand samples from fresh exposures. Carbonate carbon isotopes were prepared by drilling 1 mg powder from a fresh sample surface. About 0.4 mg powder was placed in a 10 mL Na-glass vial, sealed with a butyl rubber septum, and reacted with 100% phosphoric acid at 72 °C after flushing with helium. The evolved $CO_2$ gas was analysed for $\delta^{13}C$ using a MAT 253 mass-spectrometer coupled directly to a Finnigan Gasbench II interface (Thermo Scientific) at the State Key Lab of Biogeology and Environmental Geology (BGEG) in China University of Geosciences (Wuhan). The carbon isotopic compositions ($\delta^{13}C$) are presented as per mile (‰) relative to the Vienna Pee Dee Belemnite (V-PDB) standard. Analytical precision was better than 0.1‰, as monitored by replicate analyses of two laboratory standards (GBW 04416 and GBW 04417). Total iron concentrations were analysed by an XRF-1800 (Shimadzu Sequential X-Ray Fluorescence Spectrometer) at the State Key Lab of BGEG. Results were calibrated using two laboratory standards (GBW07105 and GBW07109). Reproducibility monitored by replicate analyses of standards and unknown samples was better than 95%. $Fe^{2+}$ contents were determined using a titration method[55]. For each sample, 0.5 g of power was dissolved in a hot 1:1 sulphuric acid. Potassium dichromate ($K_2Cr_2O_7$) and diphenylamine sulphonic acid sodium salt were used as titrant and indicator, respectively. $Fe^{3+}$ contents were calculated based on the difference between the $Fe^{2+}$ contents and total iron concentrations.

**Thermodynamic model for the formation of MRB**. In oxic oceans, iron concentration is extremely low and most dissolved iron is bound to organic molecules as Fe$L$ with a mean maximum Fe$L$ concentration of ~4 nM[43]. Considering that at higher dissolved $Fe^{2+}$ concentrations, Fe$L$ may be also higher (equilibrium process)[44], the minimum requirement for the formation of authigenic hematite is > 4 nM. Eh–$Fe^{2+}$ concentration diagram and the formation of MRB were generated based on the following equations (Standard potential and Gibbs free energy of the reactions are from ref. [56]):

$$4Fe^{2+} + 3O_2 + 6H_2O \rightarrow 4Fe(OH)_3$$

$$Fe^{2+} + 3H_2O \rightarrow Fe(OH)_3 + 3H^+ + e^-, \ Eh = 0.975 - 0.178pH - 0.059\log mFe^{2+}$$

$$FeS_2 + 8H_2O \rightarrow Fe^{2+} + 2SO_4^{2-} + 16H^+ + 14e^-, \ Eh = 0.368 - 0.068pH + 0.004\log mFe^{2+} + 0.008\log mSO_4^{2-}$$

$$2FeS_2 + 22H_2O \rightarrow 2Fe(OH)_3 + 4SO_4^{2-} + 38H^+$$

$$2Fe(OH)_3 \rightarrow Fe_2O_3 + 3H_2O$$

**Thermodynamic model for the formation of BIF**. Eh–$Fe^{2+}$ concentration diagram and the formation of BIF were generated based on the following equations (Standard potential and Gibbs free energy of the reactions are from ref [56]):

$$Fe^{2+} + 3H_2O \rightarrow Fe(OH)_3 + 3H^+ + e^-, \ Eh = 0.975 - 0.178pH - 0.059\log mFe^{2+}$$

$$FeS_2 + 8H_2O \rightarrow Fe^{2+} + 2SO_4^{2-} + 16H^+ + 14e^-, \ Eh = 0.368 - 0.068pH + 0.004\log mFe^{2+} + 0.008\log mSO_4^{2-}$$

$$12Fe(OH)_3 + CH_2O \rightarrow 4Fe_3O_4 + 19H_2O + CO_2$$

$$2Fe(OH)_3 \rightarrow Fe_2O_3 + 3H_2O$$

$$4Fe(OH)_3 + CH_2O + 3HCO_3^- \rightarrow 4FeCO_3 + 3OH^- + 7H_2O$$

**Data availability**. All data are provided in the Supplementary Information.

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

## Acknowledgements

We thank Z. Qiu, E. Jia, and Y. Ke for laboratory assistance, H. Yin, X. Hu, H. Dong, S. Zhang, X. Shi, and S. Yuan for discussions. This study is supported by State Key R&D project of China (2016YFA0601100), the National Natural Science Foundation of China (41622207, 41530104, 41661134047), and the 111 Project (B08030). This study is a contribution to the international IMBER project.

## Author contributions

Ha.S., G.J., and P.B.W. conceived the study. Ha.S., G.J., Hu.S., Z.A., D.C., L.T., and Z.S. collected samples, Ha.S., Hu.S., and D.C. completed geochemical measurements, Ha.S., G.J., S.W.P., P.B.W., J.T., and C.W. participated in discussion and interpretation. Ha.S. wrote the paper with input from all co-authors.

## Additional information

**Competing interests:** The authors declare no competing financial interests

