## [Peer Review File · Nature Communications]

Reviewers' comments:

Reviewer #1 (Remarks to the Author):

Review of Song et al. 'Switch from banded iron formation to marine red bed deposition during middle Ediacaran marks a fundamental change in ocean chemistry

Song et al. discuss the distribution of marine red beds and banded iron formations through the Precambrian and Phanerozoic, and use these distributions to place constraints on the concentration of iron in seawater over Earth history. They propose that iron concentrations fell during the Ediacaran to concentrations $<50\mu\text{M}$, that were insufficient to form BIFs, but that periods when concentrations rose to $>4\mu\text{M}$ promoted the sporadic development of marine red beds.

There are two main features of the paper in my mind: first, the quantification of iron concentrations in the ocean through time, and the use of these concentrations to characterize the timing of oxygenation (and/or presumably an increase in the availability of sulphur too??). This feature is rooted entirely in the compilation of BIF and red bed distributions, and on the modelling calculations presented in figure 4.

Second, the authors propose a mechanism for explaining the occurrence of red beds throughout the Phanerozoic, specifically that upwelling of deep waters would deliver reduced Fe and remineralised CO_2 to become fixed as oxides/siderite. This argument seems to be rooted in the new carbon isotope data presented in figure 3, and, it seems to me, by analogy to the mechanism proposed for the Shuram C-isotope excursion.

Of the two features, the first is important, well-argued, and supported by the available data (although sparse). However, I don't completely agree with the second, as I think the authors have created an unnecessary oversimplification of the processes that might be driving red bed formation. They suggest that red beds form after OAEs, as the iron that became reduced during each event was stabilized in high concentrations under oxic conditions afterwards. This iron would be concentrated by upwelling, by comparison to negative CIEs presented in figure 3. However, some observations argue against their model: (i) many Mesozoic red beds formed during OAEs, not only afterwards (e.g. New Zealand during OAE-2, Hasegawa et al., 2013). (ii) The negative CIEs shown in figure 3 are selective. Not all OAEs associated with red beds have negative CIEs. OAE-2 is a positive CIE and has evidence for red bed deposition. The T-OAE is a negative excursion embedded within a broad early Jurassic positive CIE. Red beds of ammonitico-Rosso facies occur within the positive excursion in Italy and Switzerland. The Valanginian OAE also has a broad positive CIE. (iii) How do you sustain a large inventory of dissolved Fe(II) in the deep ocean during OAEs, without oxidizing it (plenty of O_2 still available during OAEs!) or fixing it as pyrite (sulphate concentrations are typically modelled as a few mM, i.e. plenty)? I think the authors have a valid point to make, but have tried to be unnecessarily reductionist in their thinking: couldn't they simply make the point that during OAEs a lot of iron will be reduced, which will lead to a redistribution of Fe(II), which will inevitably be fixed 'elsewhere?' There is no need to start invoking palaeoceanographic processes that are inevitably very difficult to back up with data. The fact that many (but not all) red beds tend to occur at the end of events might also be due to the fact that large swathes of the ocean are suddenly being re-ventilated, which will re-oxidize a lot of pyrite. This might be tested with a critical look at dScas data following such events.

In summary, I think that the paper is interesting, well-written and easy to read, and communicates its key points very clearly. However, at the moment, the argument for the formation of red beds needs tidying up. I would suggest simplifying the paper by just concentrating on the long-term record of red bed versus BIFs.

Specific comments

Lines 100-101: the Middle-Late Jurassic examples cited are not considered as OAEs.

Lines 81-90: This discussion implies a progressive increase in 'deeper' red bed facies throughout the Phanerozoic without mentioning subduction and sampling bias.

Figure 2b: The C-isotope data shown is presumably only the authors' own, so misses a huge amount of data available for the Phanerozoic events. Couldn't published data be added to this figure for all the episodes when Fe concentrations are thought to increase?

Reviewer #2 (Remarks to the Author):

Review of the paper entitled: Switch from Banded Iron Formation to Marine Red Bed deposition during middle Ediacaran marks a fundamental change in ocean chemistry
Written by Haijun Song et al.

This study mostly focuses on tracking oceanic iron concentration during episodic anoxic Ferich (ferruginous) oceans. Evidences for Fe-rich oceans are based on sedimentary lithology observation of BIF or oceanic red beds occurrences at a global scale in terms of space and time from this study and literature sections, total iron concentration and iron speciation.

The novelties of this study lie on the suggested coupling between oceanic red beds and negative carbon (carbonate) isotope excursion during repeated anoxia since 580 Ma and a thermodynamic model for iron concentration evolution through time.

The paper is well written, illustrations however could/might be improved. Comparison with previous studies and bibliography are limited to paleolatitudes, some thickness information (but not replace in a detail stratigraphic perspective), and rough age estimate.

The reading of this manuscript raises many questions that in my opinion need to be address and more deeply discuss. More importantly, the database is not robust and supporting the discussion to me (see details below).

Major points:

1/ My most important criticism is about the database used to define MRBs through time. The authors acknowledge that total iron concentration might not be enough because it shows little variations. I agree a few hematite is enough to color a rock and total iron may also includes pyrite which is not the marker of ferruginous conditions but rather sulfidic. Indeed, ferruginous has been classically arguing using Fe_{HR}/Fe_T versus Fe_{PY}/Fe_{HR} crossplot following the work of the co-author Simon Poulton. As far as no such data are given here the authors argue line 77 that Fe^{3+}/Fe^{2+} in MRBs are clearly higher! A rapid examination of the database show that this metric is only available for two early Triassic sections measured during the course of this study with only one section having the $\delta^{13}C$ measured in parallel.

As the crux of the study is correlating MRBs with $\delta^{13}C$ in a global (space and time) perspective it is very overtake to me. Is there any literature data to compare with? Finally how the authors define MRBs? I feel surprise that the authors assess MRBs using literature sections without extracting data (at least $\delta^{13}C$ are numerous), which is the necessary step for demonstrating the global character of the geochemical signals.

2/ Figure 2 gives the feeling that MRBs are always associated with OAEs during the Phanerozoic. What about Silurian marine red beds for instance?

3/ Processes described is discussed in terms of water chemistry changes – what about changes in the early/burial diagenesis in the sediment or at the sediment/bottom water interface. Interpretations of both MRBs (for instance OAE2) and $\delta^{13}C$ carbonates (for instance middle Ediacaran (Grotzinger et al., 2011; Derry, 2010), has been link to secondary processes. Paired carbon is more robust to assess changes in the global carbon cycle there is literature data and the authors may consider do some more investigations as well.

4/ Any argument based on having a large DOC pool and oxidizing it must in fact A) have the oxidizing capacity to do so, and B) will be much more reducing after that oxidation event. Is this consistent here? Moreover, the authors have to acknowledge better the ongoing debate on these mechanisms to explain low $\delta^{13}\text{C}_{\text{carb}}$ and provide some information on the kinetic constraints to grow a large DOC reservoir and oxidizing it root on the duration of the different anoxic events (and for this precise stratigraphy is needed).

I have no specific comments on the thermodynamic being not a specialist though I agree that there is likely less iron in the Phanerozoic ocean compare to the Precambrian!

Details comments:

Line 40: According to Bekker it is GIF not BIF, this nomenclature should be introduced and explain.

Line 50: What about the other localities? There are interpretations for ferruginous, euxinic or oxic environments depending on the section and the age, more details are needed in terms of precise age and locations. The authors should make a fair review and not jump for one study to the whole ocean!

Line 57: No reference of the work by Neuhuber and Wahrenbrock! (as a general comment the authors have to acknowledge previous work and interpretations – and should think of substantially reduce auto-citation)

Line 68: Iron concentrations (Fe_2O_3) in the table 1 are very few! It is labeled Ediacaran not “middle Ediacaran”, what’s the precise age? In which phases is the iron (apply for the rest of the table), I agree that it is below 4% (weight % I guess) but not iron concentration, it is reported as oxides iron concentrations.

Line 71: This rise the question developed above, how did you decide that the rocks have to be MRBs without a homogenous measurable criterion?

Line 86 to 90: It is interesting, what does it mean?

Table 2 needs more, at least lithology and a precise stratigraphic framework that allows comparison in-between literature sections and studied samples.

In the Johnnie Formation the sequence of pink limestones includes grey limestone beds that contain crystal fan pseudomorphs typical of diagenetic process – this level of details is needed and should be discussed at least in an extended and detailed supplementary materials.

Line 108: how many exceptions? As authors claim for global signal, literature compilation of isotopic signal might be expected (and should be homogenous in weight with compilation of MRBs). For instance, Spathian $\delta^{13}\text{C}_{\text{carb}}$ reports are numerous - how do they compare with the studied samples?

Line 111: The $\delta^{13}\text{C}$ signal is not negative.

Ref 33: hydrogen sulfides! Nothing comparable to ferruginous! « Moreover, iron and zinc concentrations strongly increase at the base of the Jurassic. Higher iron concentrations are clearly linked to increased pyrite burial »

Reviewer #3 (Remarks to the Author):

Review of: Song et al. “Switch from banded iron formation to marine red bed deposition during middle Ediacaran marks a fundamental change in ocean chemistry

The authors present a compilation of marine red beds (MRB) through time, paired with carbon isotopic data from associated carbonates. They propose that marine red beds represent episodes of ocean ventilation following anoxic events. Based on thermodynamic calculations, they conclude that iron concentrations were markedly lower during those brief anoxic events compared to the Archean.

The strength of this paper is the compilation of MRB occurrences and the recognition that these are commonly preceded by anoxic events and associated with carbon isotope excursions. The

arguments are compelling and add to our understanding of biogeochemical evolution. This concept in itself would be highly relevant to the broader community and deserves publication. I would, however, urge the authors to provide more detailed stratigraphic information about their samples. Ideally provide stratigraphic columns in the appendix, alongside with additional information about the exact sampling localities.

Regarding the interpretation of the carbon isotopic data, I have two concerns. In either case, the overall importance of MRB occurrences would not be undermined. But addressing these issues would make the discussion more balanced:

First, it needs to be ruled out that the negative $\delta^{13}\text{C}$ values reflect diagenetic processes. Microbial iron reduction paired with oxidation of organic matter in the sediments could have generated isotopically light HCO_3^- . If so, then the negative $\delta^{13}\text{C}$ excursions may not necessarily be related to water column processes. Diagenetic arguments are commonly invoked to explain light $\delta^{13}\text{C}$ values in BIFs (e.g. Johnson et al. 2013, *Geology*), and also the Paleoproterozoic data presented in the appendix of this study are fairly light.

Secondly, the proposed scenario of upwelling of isotopically light HCO_3^- is not entirely convincing in its current form. The model relies on the comparison to the Black Sea (ref. 42); however, the Black Sea is strongly density-stratified. In a well-mixed marine basin, DIC may have been isotopically much better mixed. If the Black Sea is used as an analogue for the generation of isotopically light DIC under anoxic conditions, then it is also not clear why this light DIC would only appear on the shore in the aftermath of the anoxic event. It could have upwelled into the surface ocean at any time. Perhaps the authors envision that the anoxic events represent episodes of water-column stratification and stagnation. If so, then this needs to be stated more explicitly.

A bigger concern is the thermodynamic model that is used to derive iron concentrations for the Archean and post-Sturtian: First, the authors need to discuss the validity of the 4nM limit for ligand formation. Iron binding to ligands is presumably an equilibrium process. If more iron is present in solution, then also more iron may be tied up in ligands (in absolute terms). Moreover, it is possible that organic binding sites were more abundant in an anoxic ocean where biomass is not remineralized as effectively (e.g. Butterfield 2009, *Geobiology*).

Secondly, Figure 4 illustrates a strong pH dependence of the calculations. The 50 μM and 100 μM thresholds for banded iron formations are probably only valid if the pH of the early Precambrian ocean was less than 7.5. However, we do currently not have any tied constraints on the pH of ancient seawater. Reference 48 does not provide these constraints. It should at least be acknowledged in the main text that the results are strongly pH dependent and uncertain.

Minor comments:

- In the title and in line 174, it would be more accurate not to call it a 'shift from BIFs to MRBs', because the two types of deposits are separated by a long gap of Fe-oxide formation in the Mesoproterozoic (ll. 160-172). Perhaps call it 'the appearance of MRBs and their relationship with BIFs' or something like that.

- Ll. 144-148: This section is confusing at first read. It sounds as if $E_h = 0$ is a firm threshold for the transition from anoxic to oxic conditions, which is of course not the case, because the E_h scale is calibrated to the H_2 -electrode with no direct link to O_2 . I assume you take this number from Figure 4, where FeS_2 forms below $E_h = -0.15$ V or so? If yes, then the value in the text should be changed to -0.15 V.

- Ll. 147-148: The limit of 50 μM appears to be derived from Figure 4c, is that correct? If so, then it depends on the sulfate concentration. That should be added to the text.

- More detail needs to be given for the titration method. Were the rocks dissolved, and if so, is it certain that the Fe retained its redox state during the dissolution? A reference to a more detailed methods paper would help.

- Ll. 218, 220, 226: Perhaps consider greigite and other intermediate Fe-sulfide phases?

- Fig. 2: The plot of Fe concentrations through time (panel c) contains a lot more wiggles than are discussed in the text. Why is Fe assumed to drop to 10nM-0.1mM between episodes of BIF deposition in the Archean and Proterozoic? And why does Fe go as high as 0.1mM during MRB deposition, although the model can only predict a lower limit? Please either expand the discussion or draw the figure in such a way that unknown thresholds are eliminated.

- Fig. 3: Please add units to the y-axes of the $\delta^{13}\text{C}$ plots, so one can get a sense of relative longevity of these excursions.

- Fig. 4: This figure appears twice in my version of the text. Why not include the Fe-ligand species in the Precambrian panels b and c? Define FeL in the figure caption.

Overall, I enjoyed reading the manuscript. The interpretation that episodic ferruginous conditions led to the deposition of MRB is convincing. But instead of relying entirely on the thermodynamic model, perhaps the total abundance of iron, its mineralogy and stratigraphic relationships could be exploited a little more to quantify systematic differences between the MRB and BIF deposition.

Eva Stüeken

Point-by-point response to reviewers' comments

[Original reviewer comments in black; responses are in blue]

Reviewers' comments:

Reviewer #1 (Remarks to the Author):

Review of Song et al. 'Switch from banded iron formation to marine red bed deposition during middle Ediacaran marks a fundamental change in ocean chemistry

Song et al. discuss the distribution of marine red beds and banded iron formations through the Precambrian and Phanerozoic, and use these distributions to place constraints on the concentration of iron in seawater over Earth history. They propose that iron concentrations fell during the Ediacaran to concentrations $<50\mu\text{M}$, that were insufficient to form BIFs, but that periods when concentrations rose to $>4\mu\text{M}$ promoted the sporadic development of marine red beds.

There are two main features of the paper in my mind: first, the quantification of iron concentrations in the ocean through time, and the use of these concentrations to characterize the timing of oxygenation (and/or presumably an increase in the availability of sulphur too??). This feature is rooted entirely in the compilation of BIF and red bed distributions, and on the modelling calculations presented in figure 4. Second, the authors propose a mechanism for explaining the occurrence of red beds throughout the Phanerozoic, specifically that upwelling of deep waters would deliver reduced Fe and remineralised CO_2 to become fixed as oxides/siderite. This argument seems to be rooted in the new carbon isotope data presented in figure 3, and, it seems to me, by analogy to the mechanism proposed for the Shuram C-isotope excursion.

Of the two features, the first is important, well-argued, and supported by the available data (although sparse). However, I don't completely agree with the second, as I think the authors have created an unnecessary oversimplification of the processes that might be driving red bed formation. They suggest that red beds form after OAEs, as the iron that became reduced during each event was stabilized in high concentrations under oxic conditions afterwards. This iron would be concentrated by upwelling, by comparison to negative CIEs presented in figure 3. However, some observations argue against their model: (i) many Mesozoic red beds formed during OAEs, not only afterwards (e.g. New Zealand during OAE-2, Hasegawa et al., 2013). (ii) The negative CIEs shown in figure 3 are selective. Not all OAEs associated with red beds have negative CIEs. OAE-2 is a positive CIE and has evidence for red bed deposition. The T-OAE is a negative excursion embedded within a broad early Jurassic positive CIE. Red beds of ammonitico-Rosso facies occur within the positive excursion in Italy and Switzerland. The Valanginian OAE also has a broad positive CIE. (iii) How do you sustain a large inventory of dissolved Fe(II) in the deep ocean during OAEs, without oxidizing it (plenty of O_2 still available during OAEs!) or fixing it as pyrite (sulphate concentrations are typically modelled as a few mM, i.e. plenty)? I

think the authors have a valid point to make, but have tried to be unnecessarily reductionist in their thinking: couldn't they simply make the point that during OAEs a lot of iron will be reduced, which will lead to a redistribution of Fe(II), which will inevitably be fixed 'elsewhere?' There is no need to start invoking palaeoceanographic processes that are inevitably very difficult to back up with data. The fact that many (but not all) red beds tend to occur at the end of events might also be due to the fact that large swathes of the ocean are suddenly being re-ventilated, which will re-oxidize a lot of pyrite. This might be tested with a critical look at dScas data following such events.

Response: We appreciate the reviewer's thoughtful and constructive comments and suggestions. We have revised the manuscript accordingly: (1) we have provided a more comprehensive Supplementary Information to account for the reviewer's queries about the Silurian red beds, the thin red beds within OAE2, and the isotope shift associated with MRBs; (2) We have added a comprehensive section in the Supplementary Information on the debates about the origin of the Shuram excursion and modified the main text to focus on the evolution of iron concentration, and (3) we have updated the data table, figures, and references in the main text and supplementary information so that we are not overly interpreting the palaeoceanographic processes related to the iron oxidation.

In summary, I think that the paper is interesting, well-written and easy to read, and communicates its key points very clearly. However, at the moment, the argument for the formation of red beds needs tidying up. I would suggest simplifying the paper by just concentrating on the long-term record of red bed versus BIFs.

Response: We thank the reviewer for his very positive comments and thoughtful suggestions. As a result, we have simplified the discussion about the isotope excursions in the main text but provided a more comprehensive supplementary information. The manuscript now focuses on the iron concentration changes from BIF to MRB.

Specific comments

Lines 100-101: the Middle-Late Jurassic examples cited are not considered as OAEs.

Response: Revised. OAEs are now specified with updated references.

Lines 81-90: This discussion implies a progressive increase in 'deeper' red bed facies throughout the Phanerozoic without mentioning subduction and sampling bias.

Response: Thanks. We have modified the text. The more abundant record of "deeper" red bed facies is likely due to better preservation of unsubducted, slope-basin sediments (see Lines 94-95).

Figure 2b: The C-isotope data shown is presumably only the authors' own, so misses a huge amount of data available for the Phanerozoic events. Couldn't published data be added to this figure for all the episodes when Fe concentrations are thought to increase?

Response: OK, we have updated C-isotope data in supplementary data Table 3 and updated Figure 2c.

Reviewer #2 (Remarks to the Author):

Review of the paper entitled: Switch from Banded Iron Formation to Marine Red Bed deposition during middle Ediacaran marks a fundamental change in ocean chemistry
Written by Haijun Song et al.

This study mostly focuses on tracking oceanic iron concentration during episodic anoxic Ferich (ferruginous) oceans. Evidences for Fe-rich oceans are based on sedimentary lithology observation of BIF or oceanic red beds occurrences at a global scale in terms of space and time from this study and literature sections, total iron concentration and iron speciation. The novelties of this study lie on the suggested coupling between oceanic red beds and negative carbon (carbonate) isotope excursion during repeated anoxia since 580 Ma and a thermodynamic model for iron concentration evolution through time.

The paper is well written, illustrations however could/might be improved. Comparison with previous studies and bibliography are limited to paleolatitudes, some thickness information (but not replace in a detail stratigraphic perspective), and rough age estimate.

Response: Thanks for the suggestions. Thickness information are added in the new version of Figure 2 (see Figure 2b). Other detailed information, e.g. thickness data, lithology, and stratigraphic unites, are added in the new Supplementary Information (see Supplementary Table 2).

The reading of this manuscript raises many questions that in my opinion need to be address and more deeply discuss. More importantly, the database is not robust and supporting the discussion to me (see details below).

Major points:

1/ My most important criticism is about the database used to define MRBs through time. The authors acknowledge that total iron concentration might not be enough because it shows little variations. I agree a few hematite is enough to color a rock and total iron may

also includes pyrite which is not the marker of ferruginous conditions but rather sulfidic. Indeed, ferruginous has been classically arguing using FeHR/FeT versus FePY/FeHr crossplot following the work of the co-author Simon Poulton. As far as no such data are given here the authors argue line 77 that Fe³⁺/Fe²⁺ in MRBs are clearly higher! A rapid examination of the database show that this metric is only available for two early Triassic sections measured during the course of this study with only one section having the d¹³C measured in parallel.

Response: We have updated the supplementary data table. We also added information about the petrographic observation in the supplementary information. More discussions are added in the new version to emphasize that the higher Fe³⁺/Fe²⁺ not only occurred in the Early Triassic MRBs but also appeared in the Cretaceous MRBs (see Wang et al., 2011 [Ref. 25]; Hu et al., 2012 [Ref. 26]). For the d¹³C data in the Early Triassic, both of the two sections have C-isotope data. Data from Tulong section are measured in this study, and the data from Guandao have been published by Payne et al. (2004, Science).

As the crux of the study is correlating MRBs with d¹³C in a global (space and time) perspective it is very overtake to me. Is there any literature data to compare with? Finally how the authors define MRBs? I feel surprise that the authors assess MRBs using literature sections without extracting data (at least d¹³C are numerous), which is the necessary step for demonstrating the global character of the geochemical signals.

Response: We appreciate the reviewer's comments. We have updated C-isotope data in Figure 2c and Supplementary Information (see Supplementary Table 3). We also added sections about the Shuram excursion in the Supplementary Information.

2/ Figure 2 gives the feeling that MRBs are always associated with OAEs during the Phanerozoic. What about Silurian marine red beds for instance?

Response: We have added the reference and discussions about the Silurian red beds in the main text and supplementary information. The Silurian red beds are mostly siliciclastic red beds surrounding uplifted "old lands" and considered as of detrital origin. We did not include this in this compilation but if future works proves it's origin as of marine precipitation or authigenic, it could be another comparable MRBs (see updated supplementary information).

3/ Processes described is discussed in terms of water chemistry changes – what about changes in the early/burial diagenesis in the sediment or at the sediment/bottom water interface. Interpretations of both MRBs (for instance OAE2) and d¹³C carbonates (for instance middle Ediacaran (Grotzinger et al., 2011; Derry, 2010), has been link to secondary processes. Paired carbon is more robust to assess changes in the global carbon cycle there is literature data and the authors may consider do some more investigations as

well.

Responses: Thanks. We have added new sections about the origin of the Shuram excursion in the supplementary information. There are many concerns about the diagenetic alteration of the primary $\delta^{13}\text{C}$ signature of the Shuram excursion, but the global occurrence of the Shuram excursion and the preservation of delicate fabrics in carbonate particles do not support a complete reset of isotope values through burial diagenesis. The more recent authigenic model (Schrag et al., 2013) explains some of the spatial variations and local variability, but has difficulties of interpreting isotope signatures down to $\leq -12\text{‰}$ (Please see the section about the debates on the Shuram excursion in Supplementary Information).

4/ Any argument based on having a large DOC pool and oxidizing it must in fact A) have the oxidizing capacity to do so, and B) will be much more reducing after that oxidation event. Is this consistent here? Moreover, the authors have to acknowledge better the ongoing debate on these mechanisms to explain low $\delta^{13}\text{C}_{\text{carb}}$ and provide some information on the kinetic constraints to grow a large DOC reservoir and oxidizing it root on the duration of the different anoxic events (and for this precise stratigraphy is needed).

Response: Thanks. We have added the discussion about the DOC reservoir in the Supplementary Information. At the same time, we have simplified the main text to focus on direct documentation and interpretation.

I have no specific comments on the thermodynamic being not a specialist though I agree that there is likely less iron in the Phanerozoic ocean compare to the Precambrian!

Details comments:

Line 40: According to Bekker it is GIF not BIF, this nomenclature should be introduced and explain.

Response: Checked, it is BIF, see Figure 12 in Bekker et al., 2010.

Line 50: What about the other localities? There are interpretations for ferruginous, euxinic or oxic environments depending on the section and the age, more details are needed in terms of precise age and locations. The authors should make a fair review and not jump for one study to the whole ocean!

Response: Revised. Detailed locality information are added (see Lines 49-51).

Line 57: No reference of the work by Neuhuber and Wagreich! (as a general comment the authors have to acknowledge previous work and interpretations – and should think of substantially reduce auto-citation)

Response: Added. New reference Neuhuber, Wagreich et al. 2007 has been added in

the new manuscript (see Ref. 24).

Line 68: Iron concentrations (Fe₂O₃) in the table 1 are very few! It is labeled Ediacaran not "middle Ediacaran", what's the precise age? In which phases is the iron (apply for the rest of the table), I agree that it is below 4% (weight % I guess) but not iron concentration, it is reported as oxides iron concentrations.

Response: Revised. Precise ages have been given in the new version (see Supplementary Table 1). The unit of iron concentration have also been added.

Line 71: This rise the question developed above, how did you decide that the rocks have to be MRBs without a homogenous measurable criterion?

Response: The MRBs are characterized by homogenous colouration under microscopes and checked under SEM. We have updated the text and included such information in the Supplementary.

Line 86 to 90: It is interesting, what does it means?

Response: It is likely due to better preservation of unsubducted deep-water facies in general. We have added the explanation in the revised manuscript (see Lines 94-95).

Table 2 needs more, at least lithology and a precise stratigraphic framework that allows comparison in-between literature sections and studied samples. In the Johnnie Formation the sequence of pink limestones includes grey limestone beds that contain crystal fan pseudomorphs typical of diagenetic process – this level of details is needed and should be discuss at least in an extended and detail supplementary materials.

Response: More information including lithology and precise stratigraphic framework are added in Supplementary Table 2. The discussion about the crystal fan and other features of the Johnnie Formation and other middle Ediacaran MRBs have been added in the Supplementary Information.

Line 108: how many exceptions? As authors claim for global signal, literature compilation of isotopic signal might be expected (and should be homogenous in weight with compilation of MRBs). For instance, Spathian d¹³C carb reports are numerous - how do they compare with the studied samples?

Response: Revised. More published C-isotope data have been added in the new manuscript (see Figure 2c and Supplementary Table 3).

Line 111: The d¹³C signal is not negative.

Response: Revised. "negative" has been removed in the new manuscript. Instead, we state that the isotope values are lower than in adjacent units.

Ref 33: hydrogen sulfides! Nothing comparable to ferruginous! « Moreover, iron and zinc concentrations strongly increase at the base of the Jurassic. Higher iron concentrations are clearly linked to increased pyrite burial »

Response: Revised. Ref. 33 (Ricoz et al., 2012) has been replaced by Schootbrugge et al., 2013 (see Ref. 40).

Reviewer #3 (Remarks to the Author):

Review of: Song et al. "Switch from banded iron formation to marine red bed deposition during middle Ediacaran marks a fundamental change in ocean chemistry

The authors present a compilation of marine red beds (MRB) through time, paired with carbon isotopic data from associated carbonates. They propose that marine red beds represent episodes of ocean ventilation following anoxic events. Based on thermodynamic calculations, they conclude that iron concentrations were markedly lower during those brief anoxic events compared to the Archean.

The strength of this paper is the compilation of MRB occurrences and the recognition that these are commonly preceded by anoxic events and associated with carbon isotope excursions. The arguments are compelling and add to our understanding of biogeochemical evolution. This concept in itself would be highly relevant to the broader community and deserves publication. I would, however, urge the authors to provide more detailed stratigraphic information about their samples. Ideally provide stratigraphic columns in the appendix, alongside with additional information about the exact sampling localities.

Response: We appreciate the reviewer's supportive and constructive comments. We have updated the stratigraphic information including stratigraphic columns and exact sampling localities in the Supplementary Information (see Supplementary Table 1 and Figures 3-7).

Regarding the interpretation of the carbon isotopic data, I have two concerns. In either case, the overall importance of MRB occurrences would not be undermined. But addressing these issues would make the discussion more balanced: First, it needs to be ruled out that the negative $\delta^{13}\text{C}$ values reflect diagenetic processes. Microbial iron reduction paired with oxidation of organic matter in the sediments could have generated isotopically light HCO_3^- .

If so, then the negative $\delta^{13}\text{C}$ excursions may not necessarily be related to water column processes. Diagenetic arguments are commonly invoked to explain light $\delta^{13}\text{C}$ values in BIFs (e.g. Johnson et al. 2013, *Geology*), and also the Paleoproterozoic data presented in the appendix of this study are fairly light.

Response: Many thanks! We have added a comprehensive review about the debates on the origin of the Shuram excursion in the Supplementary Information. We have modified the text accordingly and added additional $\delta^{13}\text{C}$ data in Supplementary Table 3.

Secondly, the proposed scenario of upwelling of isotopically light HCO_3^- is not entirely convincing in its current form. The model relies on the comparison to the Black Sea (ref. 42); however, the Black Sea is strongly density-stratified. In a well-mixed marine basin, DIC may have been isotopically much better mixed. If the Black Sea is used as an analogue for the generation of isotopically light DIC under anoxic conditions, then it is also not clear why this light DIC would only appear on the shore in the aftermath of the anoxic event. It could have upwelled into the surface ocean at any time. Perhaps the authors envision that the anoxic events represent episodes of water-column stratification and stagnation. If so, then this needs to be stated more explicitly.

Response: Thanks. We agree with the reviewer that the negative ^{13}C excursion is not the major focus of this paper and also the origin of the Shuram event is much more complicated than just an upwelling episode. Therefore, we have simplified the discussion about excursions and move forward to our major focus, i.e. the iron concentration of ferruginous water. As noted above, we have added a section to the Supplementary Information that discusses the origin of the Shuram excursion. We also emphasize that our data contribute to the debate and is consistent with the transfer of ^{13}C -depleted carbon into carbonates, but it cannot fully resolve this issue.

A bigger concern is the thermodynamic model that is used to derive iron concentrations for the Archean and post-Sturtian: First, the authors need to discuss the validity of the 4nM limit for ligand formation. Iron binding to ligands is presumably an equilibrium process. If more iron is present in solution, then also more iron may be tied up in ligands (in absolute terms). Moreover, it is possible that organic binding sites were more abundant in an anoxic ocean where biomass is not remineralized as effectively (e.g. Butterfield 2009, *Geobiology*).

Response: Thanks. Discussions about iron-binding ligands have been added. We agree with the reviewer that iron binding to ligands is an equilibrium process. More iron may be tied up in ligands when more iron is present in solution. Relevant discussions on this point have been (see Lines 143-146).

Secondly, Figure 4 illustrates a strong pH dependence of the calculations. The 50 μM and 100 μM thresholds for banded iron formations are probably only valid if the pH of the early Precambrian ocean was less than 7.5. However, we do currently not have any tied

constraints on the pH of ancient seawater. Reference 48 does not provide these constraints. It should at least be acknowledged in the main text that the results are strongly pH dependent and uncertain.

Response: Revised. The pH values of ancient seawater are consistent with the recent estimates of ~7.5–9 for Phanerozoic and ~6.5–7.0 for Archean and Proterozoic (Halevy and Bachan, 2017). The new estimates have been added in the new manuscript.

Minor comments:

- In the title and in line 174, it would be more accurate not to call it a 'shift from BIFs to MRBs', because the two types of deposits are separated by a long gap of Fe-oxide formation in the Mesoproterozoic (ll. 160-172). Perhaps call it 'the appearance of MRBs and their relationship with BIFs' or something like that.

Response: Thanks. We have changed the title to "the onset of widespread marine red beds and the evolution of the ferruginous oceanic conditions in Earth history". We also changed the sentence of this paragraph accordingly.

- Ll. 144-148: This section is confusing at first read. It sounds as if $E_h = 0$ is a firm threshold for the transition from anoxic to oxic conditions, which is of course not the case, because the Eh scale is calibrated to the H₂-electrode with no direct link to O₂. I assume you take this number from Figure 4, where FeS₂ forms below $E_h = -0.15$ V or so? If yes, then the value in the text should be changed to -0.15 V.

Response: Thanks. To make it more precise, we have specified it as $E_h < 0$.

- Ll. 147-148: The limit of 50 μM appears to be derived from Figure 4c, is that correct? If so, then it depends on the sulfate concentration. That should be added to the text.

Response: Yes, the limit of 50 μM depends on the sulfate concentration. But this has already been documented in the text "When iron concentrations are below this lower limit (50 μM), iron is easily exhausted by reaction with dissolved sulphide produced by sulphate reducing bacteria, even under the much lower sulphate concentrations of <100 μM estimated for Archean oceans" (see Lines 158-160).

- More detail needs to be given for the titration method. Were the rocks dissolved, and if so, is it certain that the Fe retained its redox state during the dissolution? A reference to a more detailed methods paper would help.

Response: Thanks. More detailed method as well as a reference have now been added

in the revised version (see Lines 213-215).

- Ll. 218, 220, 226: Perhaps consider greigite and other intermediate Fe-sulfide phases?

Response: Yes, greigite or other intermediate Fe-sulfide phases can appear in marine settings, but pyrite is the most common Fe-sulfide mineral in the marine settings as well as the sedimentary rocks.

- Fig. 2: The plot of Fe concentrations through time (panel c) contains a lot more wiggles than are discussed in the text. Why is Fe assumed to drop to 10nM-0.1mM between episodes of BIF deposition in the Archean and Proterozoic? And why does Fe go as high as 0.1mM during MRB deposition, although the model can only predict a lower limit? Please either expand the discussion or draw the figure in such a way that unknown thresholds are eliminated.

Response: The upper limit of iron concentration between episodes of BIF deposition in the Archean and Proterozoic depends on the lower limit of iron concentration that is needed for the deposition of BIFs. The lower limit of iron concentration during these periods is not clear but would be higher than modern oxic oceans. The upper limit of iron concentration of MRB deposition depends on the lower limit of BIF depositions.

- Fig. 3: Please add units to the y-axes of the $\delta^{13}\text{C}$ plots, so one can get a sense of relative longevity of these excursions.

Response: Added. Thickness information has been added in the y-axes of the $\delta^{13}\text{C}$ plots (see Figure 3).

- Fig. 4: This figure appears twice in my version of the text. Why not include the Fe-ligand species in the Precambrian panels b and c? Define FeL in the figure caption.

Response: Ligand species have now been added in the Precambrian panels b and c in the new version of Figure 4.

Overall, I enjoyed reading the manuscript. The interpretation that episodic ferruginous

conditions led to the deposition of MRB is convincing. But instead of relying entirely on the thermodynamic model, perhaps the total abundance of iron, its mineralogy and stratigraphic relationships could be exploited a little more to quantify systematic differences between the MRB and BIF deposition.

Response: We appreciate Dr. Stüeken's thoughtful comments and suggestions that helped significantly improve the manuscript.

Eva Stüeken

Reviewers' comments:

Reviewer #1 (Remarks to the Author):

Review of Song et al. 'The onset of widespread marine red beds and the evolution of ferruginous oceans'

I previously had the opportunity to review this paper and am pleased to see a revised version. I can see that the authors have addressed many of the comments raised by myself and other reviewers, and in my view the paper is now closer to being a very nice contribution to Nature Communications.

Despite this, I do still have two related reservations that preclude me from whole-heartedly recommending the paper be accepted as is.

1. New carbon-isotope excursion data (lines 105–113): More details are needed: why these sections/events? Why are they significant? In line 108, the text should more conservatively read: '...are identified in all newly-studied MRB intervals...' I know this is a bit pedantic, but with the current phrasing, this section does sound a bit weak.
2. Mechanism of red-bed formation: The authors currently attribute the formation of red beds to oxidation of reduced iron following episodes of expanded anoxia (note, these are not all considered to be OAEs as is currently stated!). The relationship of the red beds to negative C-isotope shifts is used as evidence to support recycling of organically derived light carbon, which may be associated with reduced iron. This argument is not accurate as currently stated, and needs to be changed. Many of the Phanerozoic events are associated with either injection of isotopically light carbon into the ocean/atmosphere (not respired CO₂), or the enhanced burial of organic matter, which drives C-isotopes heavy. C-cycle recovery (i.e. a -ve shift) after the latter events is associated with burial of CO₂ in organic matter or by weathering of silicate rocks, so cannot (and should not) be uniquely attributed to recycling of respired carbon after the events, as is done here. At the very least, the role of other C-cycle processes should be noted. At best, I would recommend removing the arguments linking MRB formation to C-isotope shifts, as it simply isn't necessary. As I said last time around (and wasn't really addressed in the revision) simply stating that intervals of expanded anoxia promote the reduction of Fe, which is subsequently oxidized post-event, is perfectly sufficient.

Minor points

3. Lines 96–97: Two OAEs at the T/J boundary? I'm aware of one negative C-isotope excursion below the T/J boundary itself, and neither are considered to be 'OAEs.'
4. Line 143: mFe₂₊ needs a definition.
5. Supplementary information: the use of 'Core x' for IODP/ODP/DSDP sites is not correct terminology. These should be labelled 'Site x,' as 'core' specifically refers to each of the 10m cored/drilled intervals that together make up the Holes (A, B, C etc) for each Site.
6. DSDP Sites 105 and 367 also have early Cretaceous MRBs, but are not included in the compilation.

Reviewer #2 (Remarks to the Author):

I appreciate the many efforts made by the authors to clarify the points raised during the first round of review. I guess it is now ready for publications in Nature Comm.

This is overall a good and interesting contribution, which will find easily a broad audience.

Christophe Thomazo

Reviewer #3 (Remarks to the Author):

Review of Song et al. "The onset of widespread marine red beds and the evolution of ferruginous oceans"

The authors present a revised version of their manuscript that addresses some of the major concerns raised in the first review process. The overall conclusion of marine red beds as Phanerozoic versions of early Precambrian banded iron formations remains unchanged. As noted in my previous review, this conclusion, alongside with the database compilation is a significant advance that will be useful for the community.

Remaining points:

- The additions of stratigraphic columns for each of the measured section greatly improves the manuscript. I just have a few minor comments: Figure 3 is still lacking an indication of vertical scale, because it only shows one time point per panel. It would also be helpful to mark the section that is plotted in the left column in Figure 3 with a different colour in the corresponding map on the right. Table S1 should list the stratigraphic position of each sample.

- The discussion of Shuram excursion in the supplements nicely reviews many of the uncertainties in interpreting carbon isotopic excursions, but it doesn't quite acknowledge the observation that $\delta^{13}\text{C}$ values in BIFs are also fairly light, which is generally interpreted as a diagenetic signal. Secondly, the Shuram $\delta^{13}\text{C}$ data look more similar in magnitude to BIFs than to Phanerozoic marine red beds (Fig. 2). The statement in ll. 173 (~ carbon cycle in Ediacaran & Phanerozoic MRBs more similar to each other than to Archean & Proterozoic) therefore seems incorrect. The possibility of diagenetic oxidation of organic matter, possibly coupled to iron oxides, should be pointed out more explicitly in the main text.

- I understand that thermodynamic constraints on the importance of ligands ancient settings are very difficult to obtain. This will probably remain the weakest part of the manuscript. So I wonder if the arguments could be bolstered with independent evidence. In principle, $>4\text{nm}$ and $50\mu\text{M}$ are the same. Hence the manuscript would benefit from independent evidence that Phanerozoic anoxic waters had less iron than the Paleoproterozoic and Archean ocean. This conclusion makes a lot of intuitive sense, because BIFs are often more massive than MRBs and because it is generally thought that there was more hydrothermal activity and less sulfur in the early Precambrian. Perhaps these facts could be highlighted a bit more with perhaps a quantitative comparison of total iron oxide amounts?

- Ll. 144-145: Instead of $E_h < 0$ and $E_h > 0$, it should read $E_h < -0.18$ and $E_h > -0.18$ (or whatever the exact value is below which FeS_2 becomes abundant in Fig. 4). It is strictly speaking an assumption that E_h was less than -0.18 (i.e. low enough to form FeS_2 with sufficient sulfur, Fig. 4) in the earlier Precambrian and above -0.18 thereafter. Importantly, it is in both cases the E_h of the anoxic deep ocean. What process or ionic species would have maintained such a relatively high E_h in the anoxic deep ocean during Phanerozoic anoxic events?

Overall, the manuscript is well written and will be a useful contribution.

Eva Stüeken

Point-by-point response to reviewers' comments

[Original reviewer comments in black; responses are in blue]

Reviewers' comments:

Reviewer #1 (Remarks to the Author):

Review of Song et al. 'The onset of widespread marine red beds and the evolution of ferruginous oceans'

I previously had the opportunity to review this paper and am pleased to see a revised version. I can see that the authors have addressed many of the comments raised by myself and other reviewers, and in my view the paper is now closer to being a very nice contribution to Nature Communications. Despite this, I do still have two related reservations that preclude me from whole-heartedly recommending the paper be accepted as is.

1. New carbon-isotope excursion data (lines 105–113): More details are needed: why these sections/events? Why are they significant? In line 108, the text should more conservatively read: '...are identified in all newly-studied MRB intervals...' I know this is a bit pedantic, but with the current phrasing, this section does sound a bit weak.

Response: Thanks. We have added more details for this part. In addition, to make it more conservatively read, we revised the means of expression for the sentence '... are identified ...' following the reviewer's suggestion.

2. Mechanism of red-bed formation: The authors currently attribute the formation of red beds to oxidation of reduced iron following episodes of expanded anoxia (note, these are not all considered to be OAEs as is currently stated!). The relationship of the red beds to negative C-isotope shifts is used as evidence to support recycling of organically derived light carbon, which may be associated with reduced iron. This argument is not accurate as currently stated, and needs to be changed. Many of the Phanerozoic events are associated with either injection of isotopically light carbon into the ocean/atmosphere (not respired CO₂), or the enhanced burial of organic matter, which drives C-isotopes heavy. C-cycle recovery (i.e. a -ve shift) after the latter events is associated with burial of CO₂ in organic matter or by weathering of silicate rocks, so cannot (and should not) be uniquely attributed to recycling of respired carbon after the events, as is done here. At the very least, the role of other C-cycle processes should be noted. At best, I would recommend removing the arguments linking MRB formation to C-isotope shifts, as it simply isn't necessary. As I said last time around (and wasn't really addressed in the revision) simply stating that intervals of expanded anoxia promote the reduction of Fe, which is subsequently oxidized post-event, is perfectly sufficient.

Response: Revised. We have removed the sentences about the relationship of the red

beds to negative C-isotope shifts. Instead, we added that diagenetic processes including iron reduction may have contributed to the heterogeneity of the Shuram excursion.

Minor points

3. Lines 96–97: Two OAEs at the T/J boundary? I'm aware of one negative C-isotope excursion below the T/J boundary itself, and neither are considered to be 'OAEs.'

Response: Here is "two OAEs at the Triassic-Jurassic boundary and during the Toarcian". We added "respectively" at the end of this sentence. TJB anoxia facies are widespread in Europe (see Wignall and Hallam, 1991, Geological Society London; van de Schootbregge et al., 2013, Palaeontology).

4. Line 143: mFe²⁺ needs a definition.

Response: Added. A definition has been added in the revised manuscript.

5. Supplementary information: the use of 'Core x' for IODP/ODP/DSDP sites is not correct terminology. These should be labelled 'Site x,' as 'core' specifically refers to each of the 10m cored/drilled intervals that together make up the Holes (A, B, C etc) for each Site.

Response: Revised. Correct terminologies are used in the revised Supplementary information.

6. DSDP Sites 105 and 367 also have early Cretaceous MRBs, but are not included in the compilation.

Response: Added. DSDP Sites 105 and 367 have been added in the Supplementary information.

Reviewer #2 (Remarks to the Author):

I appreciate the many efforts made by the authors to clarify the points raised during the first round of review. I guess it is now ready for publications in Nature Comm. This is overall a good and interesting contribution, which will find easily a broad audience.

Christophe Thomazo

Response: Thanks for Dr. Thomazo's positive comments.

Reviewer #3 (Remarks to the Author):

Review of Song et al. "The onset of widespread marine red beds and the evolution of

ferruginous oceans”

The authors present a revised version of their manuscript that addresses some of the major concerns raised in the first review process. The overall conclusion of marine red beds as Phanerozoic versions of early Precambrian banded iron formations remains unchanged. As noted in my previous review, this conclusion, alongside with the database compilation is a significant advance that will be useful for the community.

Remaining points:

- The additions of stratigraphic columns for each of the measured section greatly improves the manuscript. I just have a few minor comments: Figure 3 is still lacking an indication of vertical scale, because it only shows one time point per panel. It would also be helpful to mark the section that is plotted in the left column in Figure 3 with a different colour in the corresponding map on the right. Table S1 should list the stratigraphic position of each sample.

Response: Added. Vertical bars have been added in Figure 3. Stratigraphic position of each sample has been listed in Table S1.

- The discussion of Shuram excursion in the supplements nicely reviews many of the uncertainties in interpreting carbon isotopic excursions, but it doesn't quite acknowledge the observation that $\delta^{13}\text{C}$ values in BIFs are also fairly light, which is generally interpreted as a diagenetic signal. Secondly, the Shuram $\delta^{13}\text{C}$ data look more similar in magnitude to BIFs than to Phanerozoic marine red beds (Fig. 2). The statement in ll. 173 (~ carbon cycle in Ediacaran & Phanerozoic MRBs more similar to each other than to Archean & Proterozoic) therefore seems incorrect. The possibility of diagenetic oxidation of organic matter, possibly coupled to iron oxides, should be pointed out more explicitly in the main text.

Response: Revised. The possibility of diagenetic oxidation of organic matter in BIFs and MRBs has been added in new manuscript, see Lines 115-118. In addition, the statement “iron-carbon cycle in Ediacaran and Phanerozoic MRBs are more similar to each other than to Archean and Proterozoic” has been modified to reflect the iron geochemical similarity instead of the carbon isotope values.

- I understand that thermodynamic constraints on the importance of ligands ancient settings are very difficult to obtain. This will probably remain the weakest part of the manuscript. So I wonder if the arguments could be bolstered with independent evidence. In principle, $>4\text{nm}$ and $50\mu\text{M}$ are the same. Hence the manuscript would benefit from independent evidence that Phanerozoic anoxic waters had less iron than the Paleoproterozoic and Archean ocean. This conclusion makes a lot of intuitive sense, because BIFs are often more massive than MRBs and because it is generally thought that there was more hydrothermal activity and less sulfur in the early Precambrian. Perhaps these facts could be highlighted a bit more with perhaps a quantitative comparison of total iron oxide amounts?

Response: Thanks. We have added a quantitative comparison of the thickness as well as the magnitude of iron amounts.

- Ll. 144-145: Instead of $Eh < 0$ and $Eh > 0$, it should read $Eh < -0.18$ and $Eh > -0.18$ (or whatever the exact value is below which FeS₂ becomes abundant in Fig. 4). It is strictly speaking an assumption that Eh was less than -0.18 (i.e. low enough to form FeS₂ with sufficient sulfur, Fig. 4) in the earlier Precambrian and above -0.18 thereafter. Importantly, it is in both cases the Eh of the anoxic deep ocean. What process or ionic species would have maintained such a relatively high Eh in the anoxic deep ocean during Phanerozoic anoxic events?

Response: Revised. $Eh < 0$ has been replaced by $Eh < -0.16$. For the relatively high Eh in anoxic waters during Phanerozoic anoxia events, it can be maintained by several ionic species, e.g. Fe²⁺, Mn²⁺, NH⁴⁺, NO₂⁻.

Overall, the manuscript is well written and will be a useful contribution.

Response: Thanks for Dr. Stüeken's positive comments.

Eva Stüeken

REVIEWERS' COMMENTS:

Reviewer #1 (Remarks to the Author):

I applaud the authors for making considerable efforts to amend their paper in accord with the comments of the various reviewers. I think the paper is now in a good state for publication (and is a very good contribution to the literature), but would recommend a couple of minor adjustments during the publication process:

1. Figure S1 still refers to DSDP/ODP etc Sites as 'Cores' and needs changing.
2. I'm still wary of calling every episode of anoxic facies in the Phanerozoic an 'OAE' (the T-J boundary is NOT considered as an OAE, even if anoxic facies are identified in the UK and northern Europe!). However, at this stage I'm aware I'm probably being pedantic, so I'll leave it to the editor to make a final judgement.

Reviewer #3 (Remarks to the Author):

The authors have adequately addressed all important points. I don't have any additional comments and look forward to seeing the manuscript in print.

Eva Stüeken

Point-by-point response to reviewers' comments

[Original reviewer comments in black; responses are in blue]

Reviewer #1 (Remarks to the Author):

I applaud the authors for making considerable efforts to amend their paper in accord with the comments of the various reviewers. I think the paper is now in a good state for publication (and is a very good contribution to the literature), but would recommend a couple of minor adjustments during the publication process:

1. Figure S1 still refers to DSDP/ODP etc Sites as 'Cores' and needs changing.

Response: Revised. Correct terminology are used in the Figure S1.

2. I'm still wary of calling every episode of anoxic facies in the Phanerozoic an 'OAE' (the T-J boundary is NOT considered as an OAE, even if anoxic facies are identified in the UK and northern Europe!). However, at this stage I'm aware I'm probably being pedantic, so I'll leave it to the editor to make a final judgement.

Response: Revised. OAE for T-J boundary are replaced by 'regional anoxia event' (see lines 96).

Reviewer #3 (Remarks to the Author):

The authors have adequately addressed all important points. I don't have any additional comments and look forward to seeing the manuscript in print.

Eva Stüeken